# Therapeutic Fasting: Are Patients Aged 65 and Over Ready?

**DOI:** 10.3390/nu14102001

**Published:** 2022-05-10

**Authors:** Baptiste Gramont, Martin Killian, Elodie Bernard, Laure Martinez, Sebastien Bruel, Bogdan Galusca, Nathalie Barth, Thomas Célarier

**Affiliations:** 1Department of Internal Medicine, Saint-Etienne University Hospital, CEDEX 02, 42055 Saint-Etienne, France; martin.killian@chu-st-etienne.fr; 2Team GIMAP, CIRI—Centre International de Recherche en Infectiologie, Université de Lyon, Université Jean Monnet, Université Claude Bernard Lyon 1, INSERM, U1111, Centre National de la Recherche Scientifique (CNRS), UMR530, CEDEX 02, 42023 Saint-Etienne, France; 3Department of General Practice, Université Jean Monnet, CEDEX 02, 42023 Saint-Etienne, France; elodie.bernard123@gmail.com (E.B.); bruel.sebastien@gmail.com (S.B.); 4Department of Clinical Gerontology, Saint-Etienne University Hospital, CEDEX 02, 42055 Saint-Etienne, France; laure.martinez@chu-st-etienne.fr (L.M.); thomas.celarier@chu-st-etienne.fr (T.C.); 5Eating Disorders, Addictions and Extreme Bodyweight Research Group (TAPE) EA 7423, Université Jean Monnet, CEDEX 02, 42023 Saint-Etienne, France; bogdan.galusca@chu-st-etienne.fr; 6Division of Endocrinology, Diabetes, Metabolism and Eating Disorders, Saint-Étienne University Hospital, CEDEX 02, 42055 Saint-Etienne, France; 7Chaire Santé des Ainés, Université Jean Monnet, CEDEX 02, 42023 Saint-Etienne, France; nathalie.barth@gerontopole-aura.fr; 8Gérontopôle Auvergne-Rhône-Alpes, 42100 Saint-Etienne, France

**Keywords:** therapeutic fasting, caloric restriction, elderly, geriatrics, survey, acceptance

## Abstract

While being the main potential beneficiaries of therapeutic fasting’s health benefits, the elderly are frequently thought of as being too fragile to fast. The main objective of our survey was to review the knowledge, practices, and acceptability of therapeutic fasting in subjects aged 65 years and over. From September 2020 to March 2021, an online questionnaire was sent to subjects aged 65 and over, using the mailing list of local organizations working in the field of aging. The mean age of the 290 respondents was 73.8 ± 6.5 years, 75.2% were women and 54.1% had higher education. Among the respondents, 51.7% had already fasted and 80.7% deemed therapeutic fasting interesting, 83.1% would be willing to fast if it was proven beneficial for their health, and 77.2% if it was proven to decrease the burden of chronic diseases. Subjects aged 65 to 74 years considered themselves as having the greatest physical and motivational abilities to perform therapeutic fasting. People aged 65 years, or more, are interested in therapeutic fasting and a large majority would be ready to fast if such practice was proven beneficial. These results pave the way for future clinical trials evaluating therapeutic fasting in elderly subjects.

## 1. Introduction

Fasting in human is defined as partial or total food deprivation, voluntary or forced, for a period of time. Different types of fasting are described, varying according to the intensity, duration and/or frequency of food deprivation [1]. Intensity refers to the amount and type of food and drink allowed [2]. Caloric restriction consists of reducing daily caloric intake by about 20–40%, every day for more than 7 days [3] and was associated with an increase in healthspan and lifespan in primates [4,5] and promote healthy aging in human [6]. Periodic fasting (PF) corresponds to fasting periods of 2 to 21 days and could have beneficial effects on human metabolic parameters [7]. However, the major limitation of this severe and prolonged fasting is that it must be carried out with a specialized medical follow-up. Then, intermittent fasting (IF) consists of maintaining the usual daily calorie intake, alternating with short and strict periods of food restriction [8]. Various types of IF have been studied which can be summarized into three categories: alternate-day fasting (i.e., alternation between ad libitum feeding days and fasting days), whole-day fasting (i.e., = 1–2 days of complete fasting per week plus ad libitum eating on the remaining days) and time-restricted feeding (TRF) (i.e., restriction of food intake to a time window of 8 h or less, every day). IF emerges as a safe strategy to improve longevity and lifespan in rodents and to enhance metabolic markers in humans [9,10,11,12]. Specifically, TRF improves some aspects of cardiometabolic health independently of weight loss and has anti-aging effects in humans [13,14]. Therapeutic fasting is more precisely characterized as the practice of one of these fasts for therapeutic purposes, associated or not with conventional treatment.

Difficulties in synthetizing data on the efficacy of fasting are related to the various qualitative and quantitative parameters of a diet, as well as the type of population and pathology it is intended for. As aforementioned, all the various types of fasting are most probably not equally beneficial and need to be rigorously evaluated in via clinical trials in humans. In a recent randomized trial, Liu and al. reported that TRF (between 8:00 a.m. and 4:00 p.m.) with a daily 25% caloric restriction was not more beneficial with regard to reduction in body weight or metabolic risk factors than the same daily caloric restriction alone [15].

In recent years, therapeutic fasting has gained increasing popularity, mainly due to several preclinical studies which have reported that caloric restriction was able to improve lifespan and delay the onset of age-related diseases [4,16,17]. Moreover, new approaches to fasting have emerged to improve its tolerance, such as the fasting-mimicking diet (FMD). FMD consists of a type of caloric restriction with a low-calorie, low-protein, and low-carbohydrate but relatively high-fat 4-day regimen, which causes similar changes in metabolic and immunologic pathways to water-only fasting [18,19,20]. In humans, and more precisely in the elderly, fasting could be difficult to implement and may exacerbate preexisting nutritional deficiencies. With a limited caloric restriction (10–40%) and a duration of 4 days, FMD appears to be more practical and safer than other fasting regimens [21], hence potentially applicable for elderly subjects. However, the majority of data concerning the potential positive effects on lifespan and immunity of FMD are extracted from animal studies.

The elderly could be interested in the potential advantages and therapeutic effects of fasting, but studies on subjects over 65 years old are rare, both in terms of health benefits, tolerance, and knowledge [22,23]. An explanation for the exclusion of elderly people from fasting studies could be the fear of poor tolerance and denutrition. FMD could be an answer to these concerns and opens the door to new studies.

The objective of the present study was to assess the knowledge, practices, and acceptability of therapeutic fasting, especially FMD, in subjects aged 65 years and over.

## 2. Materials and Methods

We designed a survey to study the knowledge, practices, and acceptability of therapeutic fasting in subjects aged 65 years and over. An electronic questionnaire, built by Limesurvey software (LimeSurvey GmbH), was sent by email using the mailing list of local (Loire County, France) organizations working in the field of aging whose missions are to promote and coordinate initiatives in favor of elderly people (“Office Stéphanois des Personnes Agées”, “Association Senior Autonomie”, “Centre communal d’Action Sociale”). Moreover, the survey was sent to the members of the “SYNAPSE” association, which is a patient association supporting research on aging [24]. The survey was conducted from September 2020 to March 2021. All the respondents had to be 65 years or older.

As no validated questionnaire assessing knowledge and acceptability of therapeutic fasting was readily available, we have developed a 32-question questionnaire subdivided into three parts (see Appendix A for the complete questionnaire). The questionnaire was designed by several healthcare professionals, including geriatricians, internists, family practitioners, and sociologists specializing in aging, and approved by a nutritionist. Specific questions about FMD-like were intended to explore acceptance and to identify the barriers and enablers to implement it.

In the first part, the knowledge about therapeutic fasting was evaluated globally with eight questions. The second part consisted of 18 questions and focused on the acceptability of therapeutic fasting and specifically on a medically supervised FMD-like (in our case, consisting of a 3-days reduction of 75% of caloric intakes without reducing water supply, for example one broth per meal with one bowl of rice per day as detailed in the question, see Appendix A). The third part (6 questions) collected clinical and social data as well as the self-perception of one’s health status.

All items were closed questions with defined response options, except three which included a free response option: “reason for previous fasting”, “living situation”, and “level of education”. For the question “reason for previous fasting”, several choices were available and respondents could explain their reason in a brief free text. The questions about the opinion on fasting (questions 1 to 4) and the acceptability of fasting in practice (questions 9 to 26) were evaluated in the form of a 5-point Likert scale, from “strongly agree” to “strongly disagree”.

All questions were single-choice, except for “reasons for previous fasting”, which was a multiple-choice question. No questions were mandatory.

Participation in the study was voluntary and anonymous. This study was approved by the Ethics Committee of Saint-Etienne University Hospital (under the reference number IRBN832020/CHUSTE) in July 2020.

Statistical analysis was carried out using SPSS version 23 (SPSS Inc., Chicago, IL, USA). The results were expressed as headcount, percentage, and mean ± standard deviation for quantitative variables. The χ^2^ or Fischer exact tests were used for subgroup analysis. For subgroup analysis, we considered the answer “yes, a lot” and “yes” as a single answer “yes” (point 1 and 2 of the 5-point Likert scale) and answer “no, rather not” and “no, not at all” as “no” (point 4 and 5 of the Likert scale). Statistical significance was defined as *p* < 0.05. Internal reliability was assessed using Cronbach’s alpha (α) for the acceptability section.

## 3. Results

From September 2020 to March 2021, we received 401 questionnaires, but 111 were incomplete. Hence, a total of 290 questionnaires were analyzed. The Cronbach’s α value obtained for the acceptability section was 0.899.

The mean age of the respondents was 73.8 ± 6.5 years, 218 (75.2%) were women and 72 (24.8%) were men. The extreme ages were 65 and 96 years, and the median age was 72 years.

Regarding the level of education, 157 participants (54.1%) had received higher education (graduate and undergraduate) (Table 1).

Concerning their self-perceived health status in comparison to their age-matched counterparts, 134 respondents (46.2%) reported better or much better health, 140 (48.3%) reported neither better or worse health, and 16 (5.5%) reported worse or much worse health.

### 3.1. Fasting-Related Knowledge and Practice

Two hundred and forty-two respondents (83.4%) had already heard of fasting in their life, and 150 (51.7%) declared having already fasted.

The more frequent reported reasons for fasting were: to improve mental and/or physical health for 66.7% (*n* = 100/150), a discovery objective for 33.3% (*n* = 50/150) and religious beliefs for 30.0% (*n* = 45/150). Specifically, 22.0% (33/150) of respondents had fasted to lose weight. The other reported motives were fasting out of solidarity for one person, a context of war for another, and a competitive sport context for a third.

Among the 150 people who had already fasted, 86 (57.3%, *n* = 86/150) found it easy or very easy (Figure 1).

### 3.2. Perception of Therapeutic Fasting

Regarding the perception of therapeutic fasting, 234 respondents (80.7%) declared that therapeutic fasting was an interesting topic. Only 24 (8.3%) considered therapeutic fasting to be a dangerous method (Figure 2).

Interestingly, the vast majority of the respondents considered themselves physically and/or mentally able to fast, with respectively 220 (75.9%) and 203 (70.0%) claims in that sense. Moreover, 187 (64.5%) respondents considered that their opinion regarding their ability to fast was irrespective of their age, 214 (73.8%) stated that their lifestyle was already compatible with fasting, and 215 (74.1%) agreed to adapt their lifestyle to achieve fasting.

Regarding age groups, respondents in the 65–74 age group were the most likely to think they had the physical (82.8% vs. 72.2% vs. 43.8%, *p* < 0.001) and motivational abilities (73.1% vs. 72.2% vs. 46.9%, *p* < 0.001) to fast, compared to the 75–84 age group and ≥85 age group (Table 2).

### 3.3. Acceptability of Therapeutic Fasting

Two hundred and forty-one (83.1%) respondents declared that they would be willing to perform therapeutic fasting (as FMD-like) if it was able to improve their overall health, 224 (77.2%) if it was able to decrease the burden of chronic diseases, and 219 (75.5%) if it was able to increase immunity. On the other hand, 134 (46.2%) respondents declared that they would be willing to perform therapeutic fasting if it was able to increase the effectiveness of vaccine administration.

Interestingly, respondents who had declared that they had already fasted were more inclined to fast again, compared to respondents who had never fasted, if it was proven beneficial for their overall health (97.3 vs. 67.9%, respectively, *p* < 0.001), or immunity (90.7 vs. 59.3%, *p* < 0.001) (Table 3).

Among the motivational factors which could influence the decision to perform therapeutic fasting, 206 (71.0%) respondents considered the short duration (<4 days) as a facilitating factor. The repetition (once per year) was also considered as such by 187 (64.5%) respondents. The favorable opinion of their primary care physician was considered a facilitating factor for 151 (52.1%) respondents, 190 (65.5%) considered the addition of a fasting-specific medical follow-up in the same way, and 120 (41.4%) respondents considered having one of their relatives fasting with them, likewise.

Finally, 178 (61.4%) respondents declared that they would be comfortable telling their primary care physician that they were performing therapeutic fasting.

## 4. Discussion

To our knowledge, this is the first survey focusing on fasting and specifically dedicated to subjects aged 65 years and over. This question appeared of great importance, as this population is consistently excluded from fasting studies, while the elderly population could be a privileged target for therapeutic fasting interventions.

Two-thirds of the respondents in this survey have declared that they would be willing to perform therapeutic fasting if it was able to improve their health. Moreover, it is interesting to note that half of the respondents had already fasted, and a majority considered this experience to be easy. In line with this, in a recent clinical study evaluating intermittent fasting in 10 participants (aged 65 and older), 7 had found fasting easy to perform, 8 have not experienced asthenia, and a majority declared that they would be willing to continue fasting and recommend it to a friend [25]. Moreover, side effects classically appear at the beginning of fasting and regress within the first 5 days [7].

The multiplicity of the possible beneficial effects described for fasting probably explains why more than half of the respondents in our survey considered this topic as interesting. In preclinical studies, FMD was able to improve neurodegenerative and immune-related disorders, as well as increase lifespan and delay the onset of age-related diseases as cancer [5,9,10,19]. In multiple sclerosis patients, a neurological auto-immune disease, FMD could potentially be effective in the treatment of relapsing symptoms [26]. In rodents, fasting provides protection against cancer and neurodegeneration [22] and sensitizes neurological cancer to specific treatment [27]. In older mice, it promotes hippocampal neurogenesis and improves cognitive performance [18]. Another positive effect of fasting is the reduction of inflammation and cardiovascular markers [28,29]. Other studies have also reported a decrease in immunosenescence, thanks to an increase in hematopoietic stem cell proliferation [30]. More studies are needed to explore the efficacy of fasting in humans and to define the best diet in terms of efficacy and safety.

In recent years, new approaches to therapeutic fasting have gained increasing attention, mainly due to several encouraging tolerance and safety data [31]. In fact, a large majority of the respondents had already heard about therapeutic fasting in the survey. The development of new specific fasting regimens, such as FMD, could enhance the feasibility of therapeutic fasting due its short duration (3–4 days) and the possibility to eat specific foods. In fact, in our study, approximately two-thirds of the respondents considered a brief fasting duration as a facilitating factor. Although preclinical data on FMD are very encouraging, they must be validated in humans using a well-designed, large clinical trial with a specific focus on safety [32,33]. In the elderly, another interesting fasting regimen would be TRF because of the lack of reduction in caloric intake. TRF has promising effects on diseases such as obesity and could protect against metabolic diseases even when briefly interrupted [34]. Moreover, in the elderly, TRF could have beneficial effects on cognitive disorders [35,36].

In our study, approximately three-quarters of the respondents considered that they had the physical and motivational abilities to fast with caloric reduction for three days. A relevant data of our survey is that this proportion decreased significantly after 85 years old, with less than half of the respondents who still considered themselves fit enough to perform, indicating a potential limit of application for FMD in this older age group. In fact, there is probably an age limit above which fasting becomes dangerous with an increased risk of malnutrition, such as in the very elderly or cancer patients with sarcopenia, for example [32]. However, in cohorts of patients with cancer, no malnutrition was described as a side effect of the evaluated fasting regimen with calorie restriction [31,37,38]. This risk is probably not relevant if macronutrient and micronutrient requirements were obtained. The main side effects reported in fasting studies are fatigue, headache, and dizziness [15].

There are many fears surrounding therapeutic fasting, especially from physicians [39]. In our survey, a minority of respondents considered fasting to be a dangerous method, and about half of the respondents considered the positive opinion of their primary care physician as an enabler to perform therapeutic fasting. The decision to fast seems to be influenced by the opinion of the physician [40], although the lack of recommendations makes it difficult for them to make a stand. In addition, fasting, which is a voluntary deprivation of food, seems to be in contradiction with the current vision of medicine, which emphasizes the early management of undernutrition and the need to fight against cachexia. Although the use of fasting in undernourished people is not debatable, there is still anxiety on the part of physicians to address the issue of fasting in non-denourished elderly subjects.

Food provides us with a framework and is accompanied by a strong social, cultural, and religious representation [41]. However, a large majority of the respondents have declared that they were ready to adapt their lifestyle in order to perform FMD. Fasting could have a potential physical and emotional impact, which needs to be taken into consideration by the physician to prevent complications.

Nevertheless, our study has many limitations. Because of the use of multiple channels to disseminate the online survey, we could not know the exact number of links sent, so we could not calculate the response rate. In addition, it cannot be ruled out that the subjects who responded to our survey correspond to individuals who were aware of health and therapeutic fasting issues, as we have used the mailing lists of associations supporting or participating in research on aging. The use of an online survey likely selected an educated population, as nearly three-quarters of the respondents had received higher education. We also note that 46.2% of our respondents considered themselves healthier than people of the same age. These data could be considered as a lack of representativeness of the global geriatric population. Finally, we have reported the knowledge and the expected attitudes of subjects aged 65 years and over rather than their practice. Therefore, the reported responses may reflect the politically desirable answer rather than the respondents’ actual beliefs. Moreover, in the survey, to explore the knowledge of the respondents, we used the term therapeutic fasting, which is a broad term without exact precision concerning the type of fasting, and that can be interpreted in different ways. Finally, in our questionnaire, we have not studied the eating habits of the respondents and we have not evaluated their caloric intake. Aging is associated with an increase in malnutrition [42], thus, we can hypothesize that the respondents may consider that they have already reduced their food intake and feel that fasting does not change their eating habits. However, our respondents were at less risk of malnutrition [43].

## 5. Conclusions

Our study reports a positive interest in therapeutic fasting among subjects aged 65 years and over. Contrary to popular belief, most subjects aged 65 years and over seem to be ready to perform fasting for a health purpose and consider themselves healthy enough to fast. More and more data suggest the potential beneficial effects of therapeutic fasting on the prevention of age-related diseases, making this population a target of choice. The evaluation of fasting’s safety in the oldest patients must necessarily benefit from more studies, especially for the evaluation of long-term secondary effects.

## Figures and Tables

**Figure 1 nutrients-14-02001-f001:**
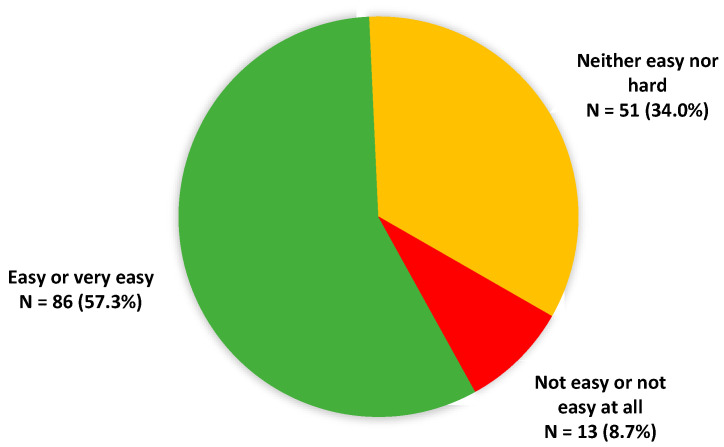
Past experience of fasting in aged respondents.

**Figure 2 nutrients-14-02001-f002:**
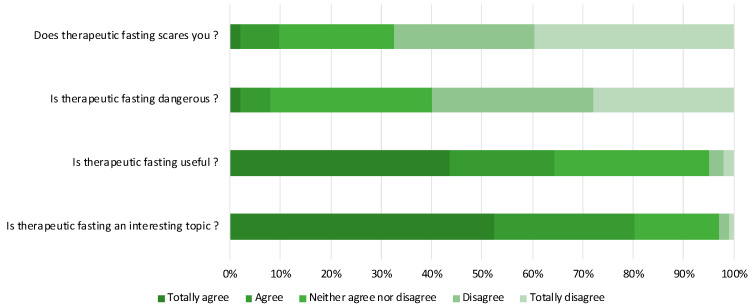
Perception of aged respondents about therapeutic fasting.

**Table 1 nutrients-14-02001-t001:** Respondents’ characteristics.

Total—*n* (%)	290 (100)
Female sex—*n* (%)	218 (75.2)
Age	
Mean (*years* ± *standard deviation*)	73.8 ± 6.5
Distribution—*n* (%)	
65 to 74	186 (64.1)
75 to 84	72 (24.8)
>84	32 (11.0)
Area—*n* (%)	
Less than 1000 inhabitants	37 (12.8)
Between 1000–10,000 inhabitants	84 (29.0)
More than 10,000 inhabitants	166 (57.2)
Unknown	3 (1.0)
Way of life—*n* (%)	
Alone	124 (42.8)
In Couple/With another family member	162 (55.9)
No answer	4 (1.4)
Educational level—*n* (%)	
No diploma	7 (2.4)
Elementary school	11 (3.8)
Junior high school	21 (7.2)
High school	94 (32.4)
Undergraduate and graduate	157 (54.1)

**Table 2 nutrients-14-02001-t002:** Relationship between age group and attitude towards fasting-mimicking diet-like (FMD-like).

	Age Group in Years—*n* (%)				
	Total	65–74	75–84	≥85	
	290 (100)	186 (64)	72 (25)	32 (11)	*p* Value
**Do you think you have the physical ability to perform a fast?**					
Yes	220 (75.9)	154 (82.8)	52 (72.2)	14 (43.8)	<0.001
Neither yes nor no	33 (11.4)	17 (9.1)	8 (11.1)	8 (25.0)	
No	37 (12.8)	15 (8.1)	12 (16.7)	10 (31.2)	
**Do you think you have the motivational ability to perform a fast?**					
Yes	203 (70.0)	136 (73.1)	52 (72.2)	15 (46.9)	<0.001
Neither yes nor no	43 (14.8)	31 (16.7)	6 (8.3)	6 (18.8)	
No	44 (15.2)	19 (10.2)	14 (19.4)	11 (34.4)	
**Do you think that fasting would be compatible with your lifestyle?**					
Yes	214 (73.8)	154 (82.8)	46 (63.9)	14 (43.8)	<0.001
Neither yes nor no	33 (11.4)	12 (6.5)	12 (16.7)	9 (28.1)	
No	43 (15.2)	20 (10.8)	14 (19.4)	9 (28.1)	
**Would you be willing to adjust your activities to perform a fast?**					
Yes	215 (74.1)	152 (81.7)	51 (70.8)	12 (37.5)	<0.001
Neither yes nor no	32 (11.0)	18 (9.7)	8 (11.1)	6 (18.8)	
No	43 (14.8)	16 (8.6)	13 (18.1)	14 (43.8)	

**Table 3 nutrients-14-02001-t003:** The comparison of different attitudes toward fasting-mimicking diet-like (FMD-like).

	Total *n* (%)	Already Fasted *n* (%)	Never Fasted *n* (%)	
	290 (11)	150 (52)	140 (48)	*p* Value
**Would you be willing to perform a therapeutic fasting if it was able to:**				
Improve your overall health				
Yes	241 (83.1)	146 (97.3)	95 (67.9)	<0.001
I do not know	30 (10.3)	2 (1.3)	28 (20.0)	
No	19 (6.6)	2 (1.3)	17 (12.1)	
Reduce the burden of chronic disease condition				
Yes	224 (77.2)	135 (90.0)	89 (63.6)	<0.001
I do not know	52 (17.9)	14 (9.3)	38 (27.1)	
No	14 (4.8)	1 (0.7)	13 (9.3)	
Reduce your medication burden				
Yes	197 (67.9)	117 (78.0)	80 (57.1)	<0.001
I do not know	65 (22.4)	27 (18.0)	38 (27.1)	
No	28 (9.7)	6 (4.0)	22 (15.7)	
Reduce the adverse effects of medication				
Yes	193 (66.6)	116 (77.3)	77 (55.0)	<0.001
I do not know	72 (24.8)	28 (18.7)	44 (31.4)	
No	25 (8.6)	6 (4.0)	19 (13.6)	
Improve your immunity system				
Yes	219 (75.5)	136 (90.7)	83 (59.3)	<0.001
I do not know	52 (17.9)	10 (6.7)	42 (30.0)	
No	19 (6.6)	4 (2.7)	15 (10.7)	
Improve the effectiveness of vaccines				
Yes	134 (46.2)	76 (50.7)	58 (41.4)	0.14
I do not know	122 (42.1)	61 (40.7)	61 (43.6)	
No	34 (11.7)	13 (8.7)	21 (15.0)	

## Data Availability

The data presented in this study are available on request from the corresponding author.

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
