# Peer review of "Therapeutic Fasting: Are Patients Aged 65 and Over Ready?"

_nutrients, 2022, doi:10.3390/nu14102001_

Round 1

Reviewer 1 Report

The paper addresses an interesting but underexamined topic.  The paper will make a contribution to literature, but I think it would be strengthened and clarified by attending to a few issues.

  1. The description of prior studies’ findings on fasting could use more details. In particular, it would be helpful to provide more details on the efficacy of the various types of fasting.  What specific findings have prior studies reported on the effects of each type of fasting on life expectancy and on risk of particular illnesses?  And, importantly, were these studies conducted on human populations?  While in some ways these details are not central to your study, which centers exclusively on perceptions of fasting, I’d argue that it’s still helpful to describe these findings accurately.  As written, readers may have the impression that the various types of fasting are equally beneficial and studies of them all have been fairly rigorous and involved human populations, which isn’t the case.
  2. I’d recommend incorporating more details from the survey itself into the manuscript, as I felt at times that additional information was needed to fully understand the study’s findings (e.g., the wording of actual items). For example, I wondered how “therapeutic fasting” was defined in the survey, as it’s likely to have different meanings to different participants.  Indeed, “fasting” would also have different meanings to different people.  Moreover, both would have different meanings to researchers and participants.  When I read the supplemental material (i.e., the survey), I realized that the term was not defined for participants, which introduces a limitation that should be addressed in the discussion section of the paper.  Without a common definition provided, we can’t be sure that some people, for example, defined it as any limitation of food intake while others had varying other definitions involving limits on the timing, duration, and type of food intake – or perhaps even variation on the goal of limiting food intake (e.g., “therapeutic” in what sense and to whom?)
  3. Related to the point on definitions made in #2, I had the impression from the manuscript’s introduction that the study was about fasting mimicking diets – but it’s not. Instead it’s about “therapeutic fasting” (as participants defined it for themselves).  The paper’s focus could be clarified in the introduction.
  4. The question about participants having adequate “mental resources” to allow fasting had unclear meaning. How were the researchers thinking of this concept and, importantly, how might the question have been interpreted by participants?  Were you asking about cognitive abilities in general, like cognitive processing skills?  Or perhaps memory?  And what about knowledge?  Was this question asking people’s perceptions of their knowledge (e.g., of fat, protein, and other compositions of foods), which would be essential to engage in FMD, for example.

Author Response

The paper addresses an interesting but underexamined topic.  The paper will make a contribution to literature, but I think it would be strengthened and clarified by attending to a few issues.

We thank reviewer 1 for her/his encouraging comments regarding the novelty of this study, but also for her/his suggestions which will definitely improve the manuscript. We hope the revised version of the paper will respond to all of her/his comments.

  1. The description of prior studies’ findings on fasting could use more details. In particular, it would be helpful to provide more details on the efficacy of the various types of fasting.  What specific findings have prior studies reported on the effects of each type of fasting on life expectancy and on risk of particular illnesses?  And, importantly, were these studies conducted on human populations?  While in some ways these details are not central to your study, which centers exclusively on perceptions of fasting, I’d argue that it’s still helpful to describe these findings accurately.  As written, readers may have the impression that the various types of fasting are equally beneficial and studies of them all have been fairly rigorous and involved human populations, which isn’t the case.

Thank you for your comments.

As highlighted by reviewer 1, the various types of fasting have indeed not been equally studied and we do understand why the reviewer has suggested adding details about them. Hence, we have added comments about the main effects (as suggested, on life expectancy and/or disease) which have been reported by prior studies in human or mammals:

  • Lines 37-40: “Caloric restriction consists of reducing daily caloric intake by about 20-40%, every day for more than 7 days [3], and was associated with an increase in healthspan and lifespan in primates (PMID: 19590001; PMID: 30271916) and promote healthy ageing in human (PMID: 27544442).”
  • Lines 40-43: “Periodic fasting (PF) corresponds to fasting periods of 2 to 21 days (PMID: 30601864) and could have beneficial effects on human metabolic parameters. However, the major limitation of this severe and prolonged fasting is that must be carried out with a specialized medical follow-up.”
  • Lines 43-59: “Then, intermittent fasting (IF) consists of maintaining the usual daily calorie intake, alternating with short and strict periods of food restriction (PMID: 31881139). Various types of IF have been studied which can be summarize in 3 categories: alternate day fasting (e., alternation between ad libitum feeding days and fasting days), whole-day fasting (i.e., = 1-2 days of complete fasting per week plus ad libitum eating on the remaining days) and time-restricted feeding (TRF) (i.e., restriction of food intake to a time window of 8 hours or less, every day). IF emerge as a safe strategie to improve longevity and lifespan in rodents and to enhance metabolic markers in human (PMID: 28761067; PMID: 35310455; PMID: 31525701; PMID: 34919135). Specifically, TRF improve some aspects of cardiometabolic health independently to the weight loss and have anti-ageing effects in human (PMID: 29754952; PMID: 31151228).”
  • Lines 230-232: In rodents, fasting provides protection against cancer and neurodegeneration and sensitize neurological cancer to specific treatment (PMID: 2298453)

However, we think (and hope reviewer 1 will agree) that adding more specific details may weight down the introduction. Nonetheless, we have added the following sentences (lines 62-68) to illustrate the complexity of summarizing the efficacy/potential efficacy of all the various types of fasting, including details about a trial which has been recently published in the NEJM: ”Difficulties to synthetize data on efficacy of fasting are related to the various qualitative and quantitative parameters of a diet, as well as the type of population and pathology it is intended for. As aforementioned, all the various types of fasting are most probably not equally beneficial and need to be rigorously evaluated in via clinical trials in human. In a recent randomized trial, Liu and al., reported that TRF (between 8:00 a.m. to 4:00 p.m) with a daily 25% caloric restriction was not more beneficial with regard to reduction in body weight or metabolic risk factors than the same daily caloric restriction alone (PMID: 35443107).”

We have also added a comment on FMD: “However, the majority of data concerning the potential positive effects on lifespan and immunity of FMD are extracted from animal studies.” (lines 79-81) and “In multiple sclerosis patient, a neurological auto-immune disease, FMD could potentially effective in the treatment of relapsing symptoms (PMID: 27239035).” (lines 229-230).

  1. I’d recommend incorporating more details from the survey itself into the manuscript, as I felt at times that additional information was needed to fully understand the study’s findings (e.g., the wording of actual items). For example, I wondered how “therapeutic fasting” was defined in the survey, as it’s likely to have different meanings to different participants. Indeed, “fasting” would also have different meanings to different people. Moreover, both would have different meanings to researchers and participants. When I read the supplemental material (i.e., the survey), I realized that the term was not defined for participants, which introduces a limitation that should be addressed in the discussion section of the paper. Without a common definition provided, we can’t be sure that some people, for example, defined it as any limitation of food intake while others had varying other definitions involving limits on the timing, duration, and type of food intake – or perhaps even variation on the goal of limiting food intake (e.g., “therapeutic” in what sense and to whom?).

Thanks for this comment. We answered to this comment with the comment 3 in the next paragraph.

  1. Related to the point on definitions made in #2, I had the impression from the manuscript’s introduction that the study was about fasting mimicking diets – but it’s not. Instead it’s about “therapeutic fasting” (as participants defined it for themselves).  The paper’s focus could be clarified in the introduction.

Thank you for yours comments 2 and 3. Indeed, we agree with the necessity to correctly detail the method section. In the survey (introduction section), “therapeutic fasting” was defined as “a voluntary partial or complete deprivation of food, without reduction of water consumption, for a few days, with the expected objective of improving health.” This was intended to give a broad definition of “therapeutic fasting” in order not to limit the answers focused on general knowledge. We agree with reviewer 1 that introduces a limitation to the interpretation of our study and we have added in the discussion section: “Moreover, in the survey, to explore knowledge of the respondents, we use the term therapeutic fasting witch is a broad term without exact precision concerning the type of fasting and that can be interpreted in different ways.” (lines 303-305)

For the evaluation of the acceptability of therapeutic fasting, we proposed an example of regimen consisting of a 3-days reduction of 75% of caloric intakes without reducing water supply, for example one broth per meal with one bowl of rice per day (citing in the survey). This is not a really FMD but, as in FMD, we have suggested a fasting with non complete calorie restrcition (i.e 20-40% of standard calorie intakes with low-carbohydrates and low-protein diet) for 3 days. In the revised manuscript (and table) we modified “FMD” by “FMD-like” which may be more appropriate.

  1. The question about participants having adequate “mental resources” to allow fasting had unclear meaning. How were the researchers thinking of this concept and, importantly, how might the question have been interpreted by participants?  Were you asking about cognitive abilities in general, like cognitive processing skills?  Or perhaps memory?  And what about knowledge?  Was this question asking people’s perceptions of their knowledge (e.g., of fat, protein, and other compositions of foods), which would be essential to engage in FMD, for example.

Thanks for the precious comment regarding the need to clarify “mental ressources”. In French the term “mental resources” is common, unambiguous, and widely seen  as a self-evaluated ability/will to commit to something i.e., “Therapeutic fasting? I would definitely be able to do that”. It is not really about the cognitive abilities or the knowledge one has.

The  English term “motivational ability” may be more appropriate and has replaced “mental ressources” in the manuscript and supplementary materials. We hope the reviewer will find it suitable. 

Reviewer 2 Report

Interesting study regarding therapeutic role of fasting in elderly. Here are some suggestions:

- You should expand the introduction, explaining better different types of fasting  (Time restricted feeding, alternate day fasting ..) and potential health benefits. You can use for references PMID: 26374764  and PMID: 27304506

- Line 214; : fasting may be dangerous in patients at risk of undernutrition. You need to remark that fasting may be healthy if macronutrient and micro requirements were obtained. List for which patients it may be risky to fast and what side effects there may be during a fasting regimen

- In the discussion, it is important to distinguish effects of fasting with or without calorie restriction (so when you mention any paper please explain this).

- Another interesting fasting regimen is time restricted feeding. It consists in a reduction of feeding hours (usually 8h) and an increase in fasting hours (usually 16h). Patients can do this just through skipping breakfast (late time restricted feeding) or skipping dinner (early time restricted feeding) at the same calorie intake. Among all intermittent fasting regimens, it appears to be the easiest and safest for the elderly. You can use for references PMID: 25470547 and PMID: 33356688 and PMID: 33435416 and PMID: 34100325

Author Response

Interesting study regarding therapeutic role of fasting in elderly. Here are some suggestions:

Thanks to reviewer 2 for her/his comments and interesting suggestions. We hope she/he will be satisfied with the revised version of the paper.

  1. You should expand the introduction, explaining better different types of fasting  (Time restricted feeding, alternate day fasting ..) and potential health benefits. You can use for references PMID: 26374764  and PMID: 27304506

Thanks for this comment, in line with the 1st comment from reviewer 1 which has already resulted in the addition of multiple precisions regarding the previous findings and potential health benefits of the main types of fasting (see above).

Nonetheless, we have added further precisions (lines 44-56) thanks to your comment and have included the suggested references: “Various types of IF have been studied which can be summarize in 3 categories: alternate day fasting (i.e., alternation between ad libitum feeding days and fasting days), whole-day fasting (i.e., = 1-2 days of complete fasting per week plus ad libitum eating on the remaining days) and time-restricted feeding (TRF) (i.e., restriction of food intake to a time window of 8 hours or less, every day)”.

We also included a briefly consideration in the discussion section: “More studies are needed to explore efficacy of fasting in human and define the best diet in term of efficay and safety.” (lines 236-237)

  1. Line 214; : fasting may be dangerous in patients at risk of undernutrition. You need to remark that fasting may be healthy if macronutrient and micro requirements were obtained. List for which patients it may be risky to fast and what side effects there may be during a fasting regimen.

Thank you for this important point, which indeed needs precisions. Hence, we have corrected lines 267-274 as such: “In fact, there is probably an age limit above which fasting becomes dangerous with an increased risk of malnutrition, such as very elderly or cancer patient with sarcopenia for example (PMID: 32223459). However, in cohorts of patients with cancer no malnutrition were described as a side effect for the evaluated fasting regimens (PMID: 27282289; PMID: 32576828; PMID: 26438237). This risk is probably not relevant if macronutrient and micronutrient requirements were obtained. The main side effects reported in fasting studies are fatigue, headache and dizziness (PMID: 35443107).”

  1. In the discussion, it is important to distinguish effects of fasting with or without calorie restriction (so when you mention any paper please explain this).

Thank you for your comment that underline the necessity to correctly define the presence/absence of calorie restriction. The presence or absence of calorie restriction has now been indicated where applicable.

  1. Another interesting fasting regimen is time restricted feeding. It consists in a reduction of feeding hours (usually 8h) and an increase in fasting hours (usually 16h). Patients can do this just through skipping breakfast (late time restricted feeding) or skipping dinner (early time restricted feeding) at the same calorie intake. Among all intermittent fasting regimens, it appears to be the easiest and safest for the elderly. You can use for references PMID: 25470547 and PMID: 33356688 and PMID: 33435416 and PMID: 34100325

Thank you for your comments and the references. We agree with reviewer 2 about the fact that TRF may be one of the easiest and safest fasting regimen in elderly, by maintaining caloric intake. We corrected the introduction as detailed previously and added more information about TRF in discussion section (lines 246-250): “In the elderly, another interesting fasting regimen would be TRF because of the lack of reduction in caloric intake. TRF have promising effects on diseases such as obesity and could protect against metabolic diseases evenwhen briefly interrupted (PMID: 25470547). Moreover, in elderly, TRF could have beneficial effects in cognitive disorders (PMID: 33435416; PMID: 34100325)” and have cited your references.